# miRNAs as Interconnectors between Obesity and Cancer

**DOI:** 10.3390/ncrna10020024

**Published:** 2024-04-15

**Authors:** Grecia Denisse González-Sánchez, Angelica Judith Granados-López, Yamilé López-Hernández, Mayra Judith García Robles, Jesús Adrián López

**Affiliations:** 1Doctorate in Biosciences, University Center of Los Altos, University of Guadalajara, Tepatitlán de Morelos C.P. 47620, Mexico; grecia.gonzalez7738@alumnos.udg.mx; 2Laboratory of microRNAs and Cancer, Academic Unit of Biological Sciences, Autonomous University of Zacatecas “Francisco García Salinas”, Zacatecas C.P. 98066, Mexico; agranados@uaz.edu.mx; 3Laboratory of Proteomics and Metabolomics, Cátedras-CONACYT, Academic Unit of Biological Sciences, Autonomous University of Zacatecas “Francisco García Salinas”, Zacatecas C.P. 98066, Mexico; ylopezher@conacyt.mx; 4Biotechnology Department of the Polytechnic, University of Zacatecas, Fresnillo, Zacatecas C.P. 99059, Mexico

**Keywords:** obesity, cancer, miRNAs, lncRNAs

## Abstract

Obesity and cancer are a concern of global interest. It is proven that obesity may trigger the development or progression of some types of cancer; however, the connection by non-coding RNAs has not been totally explored. In the present review, we discuss miRNAs and lncRNAs dysregulation involved in obesity and some cancers, shedding light on how these conditions may exacerbate one another through the dysregulation of ncRNAs. lncRNAs have been reported as regulating microRNAs. An in silico investigation of lncRNA and miRNA interplay is presented. Our investigation revealed 44 upregulated and 49 downregulated lncRNAs in obesity and cancer, respectively. miR-375, miR-494-3p, miR-1908, and miR-196 were found interacting with 1, 4, 4 and 4 lncRNAs, respectively, which are involved in PPARγ cell signaling regulation. Additionally, miR-130 was found to be downregulated in obesity and reported as modulating 5 lncRNAs controlling PPARγ cell signaling. Similarly, miR-128-3p and miR-143 were found to be downregulated in obesity and cancer, interacting with 5 and 4 lncRNAs, respectively, associated with MAPK cell signaling modulation. The delicate balance between miRNA and lncRNA expression emerges as a critical determinant in the development of obesity-associated cancers, presenting these molecules as promising biomarkers. However, additional and deeper studies are needed to reach solid conclusions about obesity and cancer connection by ncRNAs.

## 1. Introduction

Overweight and obesity are defined as abnormal or excessive fat accumulation that may compromise human body health. Prevalence of obesity at the global level has tripled from 1975 to 2016. According to the World Health Organization, in 2016, more than 1.9 billion adults, 18 years and older, were overweight with over 650 million classified as obese. Alarmingly, this issue extends beyond adults, with a sharp increase in overweight and obesity among children and adolescents aged 5–19, rising from 4% in 1975 to 18% in 2016 [1]. On the other hand, 18.1 million new cancer cases and 9.6 million cancer deaths occurred in 2018 worldwide. In men, lung, prostate, stomach, and non-melanoma skin cancers were the most frequently diagnosed, while in women, were the breast, lung, cervix, and colon carcinomas. The lung, liver, stomach, and prostate cancers are responsible for the majority of cancer deaths in men, and in women, the breast, lung, cervix, and stomach malignant diseases [2].

Obesity is associated with numerous health implications, encompassing diabetes mellitus, cardiovascular diseases, dyslipidemia, high blood pressure, musculoskeletal disorders, neurodegenerative diseases, and some types of cancer [3]. Epidemiological evidences suggest that a higher body mass index (BMI) increases the risk of several types of cancer, including colon, rectum, endometrium (lining of the uterus), breast (postmenopausal women), ovary, esophagus, kidney, liver, pancreas, and gallbladder, among others [4]. Furthermore, individuals affected by obesity often suffer from metabolic syndrome, a condition that can contribute in an additive or synergistic manner to the development and progression of cancer [5]. The intricate relationship between obesity and tumorigenesis involves several mechanisms, including chronic hyperinsulinemia, increase in the bioavailability of steroid hormones, localized inflammation, and a deregulated secretion of adipokines by adipocytes [6].

An obesogenic environment, along with genetic predisposition and epigenetic mechanisms, plays a crucial role in the onset of obesity [7,8]. Regarding genetic predisposition, a large meta-analysis identified 97 genome-wide association (GWS) loci associated with BMI, suggesting that common variation accounts for >20% of aberrant BMI [9].

The definition of epigenetics has changed over the years. In a general way, epigenetics refers to inheritable changes in gene expression, not related to alterations in the DNA sequence [10]. Environmental factors can trigger various epigenetic mechanisms like DNA methylation, histone modifications, and noncoding ribonucleic acid regulation, including microRNAs (miRNAs) and long non-coding RNAs (lncRNAs) [10]. Numerous studies have suggested a possible connection between miRNAs or lncRNAs in obesity, metabolic diseases, and some types of cancer [11,12,13,14]. The lncRNA MIST was downregulated in response to proinflammatory stimuli, exhibiting an inverse correlation with obesity and insulin resistance [15] while lincIRS2 was downregulated in induced obesity mediated by the transcription factor MAFG [16]. On the other hand, the expression of long intergenic noncoding H19 was reduced in the brown adipose tissue of obese mice and showed a negative correlation with BMI in human subjects [17]. In contrast, the upregulation of Meg3 expression in preadipocytes led to proliferation, adipogenesis, and angiogenesis [18]. Plnc1 has been reported to be abundantly expressed in adipose tissue where it induces the expression of PPAR-γ2, promoting adipogenesis [19]. Several lncRNAs such as lncRNA-p5549, H19, GAS5, and SNHG9 have been shown downregulated in adipose tissues, while Blnc1, AC092834.1, TINCR, and PVT1 are reported upregulated in obese subjects [20]. The association of some of these lncRNAs with cancer suggests a potential link between obesity and cancer. The augmented expression of H19 and MMP14 has been linked to colorectal cancer [21]. A lower expression of GAS5 has been observed in colorectal cancer [22]. The alteration of lncRNAs in obesity and cancer is well demonstrated; however, the link between them remains unclear. The present review firstly discusses the connection of an obesogenic environment with the development and progression of cancer on the basis of coding genes dysregulation. The second section covers the participation of microRNAs in the obesity–cancer interplay. The third part addresses the regulation of miRNAs-mRNA axis by lncRNAs, and finally, the connection between these molecules, obesity, and cancer is discussed. This review firstly discusses the role of miRNAs as part of the physiology of adipose tissue and how the deregulation of this important molecules contributes to an obesogenic environment that promotes the development and progression of obesity-related cancer. In a second part, the participation of lncRNAs in the regulation of the miRNA–mRNA axis in obesity is shown. In the second section, the regulation of lncRNAs by miRNAs is addressed. Finally, the connection between these molecules, obesity, and cancer is discussed.

## 2. Adipogenesis Linked to Cancer Development

The adipose tissue (AT) represents more than just a store of fat and excess energy; it is a complex organ, metabolically active and with endocrine function [23,24]. The composition of the AT is not limited only to adipocytes. A variety of cell types such as immune cells, endothelial cells, fibroblasts, among others, are interconnected, sharing soluble mediators like adipokines, miRNAs, among others, with autocrine/paracrine action, in order to participate collectively in metabolic homeostasis [25,26], adipogenesis, lipogenesis, and lipolysis [27,28].

Adipogenesis is a complex process in which pluripotent mesenchymal stem cells (MSCs) differentiate into adipocytes [29,30,31,32]. The bone morphogenic protein (BMP) family of ligands plays a crucial role in embryonic development and in maintaining the adult homeostasis. Their multiple actions include the regulation of cell lineage commitment, morphogenesis, differentiation, and proliferation, among others [33]. Tang et al. showed that treatment of C3H10T1/2 cells with BMP4 during proliferation and subsequent administration of appropriate inducers led to differentiation of the cells into adipocytes [34]. Similarly, experiments with 10T1/2 cells showed that treatment of the cell line with exogenous BMP4 makes these cells susceptible to differentiation inducers and their commitment to the adipocyte lineage. On the other hand, members of the Wnt family work as negative regulators of adipogenesis, as they suppress the expression of adipogenic genes to promote an undifferenced state [35]. Autocrine and paracrine communication as well as morphological changes and gene expression result in the transformation of preadipocytes into mature adipocytes [36,37,38]. MSCs derived from adipose tissue are committed to producing adipocytes by adipogenic inducers and mediators of the BMP family members [39] and inhibited by members of the Wnt family [37,40]. After exposure to inducers of differentiation (insulin-like growth factor 1 (IGF1), glucocorticoid, cyclic AMP (cAMP), and KLFs), preadipocytes enter a mitotic clonal expansion, and a switch in transcription factors and cell cycle regulators essential for the differentiation of preadipocytes, such as CCAAT/enhancer binding protein a (C/EBP-α) and peroxisome proliferator-activated receptor gamma (PPAR-γ), leads to expression of adipocyte-specific genes [29,40]. The terminal differentiation stage requires a cell cycle arrest, PPAR-γ, C/EBP-α expression, and the transcriptional activation of those genes responsible for the phenotype of the adipocyte (glucose uptake, sensitivity to insulin, fat accumulation, adipokines) [41]. In the context of cancer induction or progression, MSCs can promote epithelial mesenchymal transformation, inducing cell migration, augmenting MMP9 protein expression involved cellular matrix degradation [42] via AMPK signaling pathway in non-small-cell lung cancer (NSCLC) cells [43]. MSCs promote autophagy in NSCLC by the increase and decrease in the LC3A/B and P62 proteins expression, respectively, through AMPK phosphorylation in Thr172 [43]. The AMPK signaling pathway triggered in NSCLC by MSCs inhibits apoptosis by Bcl-xl augment and Bak protein expression decrease [43]. Additionally, MSCs can generate an immunosuppressive environment by the release of inflammatory cytokines favoring cancer development [44].

Adipose tissue represents an organ with expansion capabilities in the organism, since it is of mesenchymal origin; its distribution is divided into white and brown adipocytes, each with its own characteristics. Within this cellular differentiation, different miRNAs are involved, due to their high capacity to regulate gene expression [45]. The process of adipogenic differentiation involves a series of events involving modulators such as transcription factors, as well as biologically important regulators such as ncRNAs, which are also implicated by their influence on adipose-derived stem cells (A-DSCs) through the induction of their inhibition or overexpression in cancer development [46,47]. The connection between obesity and cancer development has been supported by the expression or activities of various molecules and metabolic and cell signaling pathways. In this regard, adipogenic differentiation and cancer development have been suggested by downregulation of catenin beta interacting protein 1 CTNNBIP1 modulating Wnt cell signaling. Additionally, adipogenic differentiation can be modulated in osteogenic differentiation through Smurf2 and PPARγ in bone marrow mesenchymal stem cells (BMSCs), two proteins involved in tumorigenesis [48]. In agreement, the inhibition of *Pparg* and *Sertad2* genes was found to constrain preadipocyte differentiation and favor lipolysis [49]. PPARγ role has been suggested as an important factor in non-small-cell lung cancer development, where this protein increase converted M1 macrophages to M2 phenotype, maintaining tumor cells out of the reach of T lymphocytes and natural killer cells (NK) [50]. In a general form of dynamic change, BMSCs increments, growth factors, and accessory cells are all needed to maintain a tumor-favorable microenvironment.

Another process that accompanies both adipogenesis and cancer is lipid droplet (LD) formation. Therefore, misregulation of adiposis molecules may trigger or improve cancer development and is an open door to explore cancer mechanisms in order to find therapeutic targets for both obesity and cancer [51,52]. In this context, certain proteins have been identified as shared by both diseases, such as Paxs, PPARs [53], and proteins of DMTF1 pathway [51,52]. Notably, cell division protein kinase 6 (CDK6) plays an active role in cell proliferation and differentiation in cancer. Interestingly, Hou et al. showed CDK6 participation as a negative regulator in the transition to brown adipose tissue [54]. Furthermore, it has been revealed that CDK6 expression mediates adipogenesis and cancer development through NEDD9, a protein involved in cell invasion, migration, and proliferation [55].

Some of the main physiological functions of adipose tissue include the capacity to differentiate into several subtypes, such as white adipose tissue, which is characterized by its endocrine function by modulating caloric intake and triglyceride storage through the formation of the lipid droplet, and brown adipose tissue, which is responsible for thermogenic processes through the oxidation of fats and glucose [56]. Heat in brown and beige adipocytes is generated by the involvement of mitochondria through PPARγ coactivator1-α (PGC1-α) [57]. Additionally, a study performed in human multipotent adipose-derived stem (hMADS) cells, showed that adipocyte differentiation inhibition is triggered by blocking transcription of the adipogenesis master genes PPARγ and C/EBPα [58]. On the other hand, brown adipocytes are specialized in the dissipation of energy in the form of heat, a function attributed to the high expression of UCP1 modulated by Zfp516, Dio2, and Ppargc1a as transcriptional regulatory genes. Also accounting for obesity-mediated cancer, the production of smaller adipocytes and lipogenesis has been associated with increased angiogenesis, causing hypoxic stress in fat reservoirs and promoting an inflammatory state [59] conducive to carcinogenesis [60].

## 3. Biogenesis of MicroRNAs

miRNAs are endogenous short non-coding RNA molecules (20–22 nucleotides) that regulate gene expression at the post-transcriptional level [61,62] and, under certain conditions, even activate or regulate translation [63]. In this context, miRNAs are considered regulatory molecules of great importance in nematodes, insects, plants, and mammals [53,62,64,65,66]. In animals, most miRNAs are transcribed by RNA polymerase II into long primary miRNAs (pri-miRNAs). Pri-miRNAs contain a stem–loop structure hosting mature miRNAs sequences [67,68,69]. As part of the initial process of maturation, the microprocessor, formed by nuclear enzyme Drosha and the double-stranded RNA (dsRNA) binding protein DGCR8 (in humans), cuts the stem-loop, producing an RNA intermediary of ~65-nt, named pre-miRNA. The incorporation of Ran-GTPase Exportin 5 carries out the pre-miRNA to the cytoplasm, where subsequently Dicer, a RNase III type enzyme, and TRBP (trans-activation response RNA-binding protein), cleaves the loop terminal, generating a small RNA duplex of ~19–24-bp [67,69,70]. The miRNA duplex is subsequently integrated with the argonaute (AGO) proteins in an-ATP-dependent event and with the participation of HSP70/HSP90 chaperones, forming a complex known as pre-RISC (RNA-induced silencing complex). The maturation of RISC involves the separation of the passenger strand from miRNA duplex and retention of guide strand [67,70,71]. Mature RISC, along with guide strand, recognizes a specific mRNA sequence through complementary base-pairing, and subsequently, post-transcriptional repression of target mRNAs is carried out [72]. In vitro and in vivo studies have clearly shown significant participation of miRNAs in essential biological functions, including cell differentiation and development, nervous system regulation, angiogenesis, apoptosis, signal transduction, organ development, immunity, and adipogenesis [73,74,75]. In this context, a miRNA dysregulation is often associated with human diseases, such as neurodevelopmental disorders, immune response pathological conditions, obesity, metabolic disorders, and cancer [66,76,77,78].

## 4. miRNAs Involved in Adipogenesis

Obesity is currently considered a serious problem for public health, because of its high rate of prevalence worldwide; this pathology is the result of the sum of several social, cultural, and biological factors [79]. In chronic diseases, there is a reduction in cellular metabolic processes such as glycolysis, beta-oxidation of fatty acids, among others; therefore, alterations in the metabolic homeostasis of adipose tissue predisposes to the development of clinical conditions such as diabetes, vascular disorders, obesity, and some types of cancer [80].

Recent studies have elucidated the multifaceted role of miRNAs in adipogenesis and morphological changes within adipocytes, as they modulate the differentiation of these cells from the first lineage stage in mesenchymal stem cells. These findings highlight the importance of how miRNAs’ misregulation can influence the dysfunction of fat tissue and promote metabolic alterations. Understanding how miRNAs modulate processes such as hypertrophy (enlargement of existing fat cells) or hyperplasia (formation of new fat cells by differentiation from resident precursors of fat cells) is crucial for maintaining adipose tissue homeostasis. Hyperplasia, in particular, contributes to adipocyte hypoxia, thus triggering a number of inflammatory tissue problems [27,81].

Several studies have identified the involvement of different miRNAs in AT and plasticity at all stages of differentiation (Table 1). At present, the participation of about 40 miRNAs present in preadipocytes have been demonstrated as well as an increasing number of circulating miRNAs involved in metabolic alterations in AT [82,83]. A report by Xi et al. showed an increased expression of miR-214-3p, which promoted adipogenic differentiation capacity in 3T3-L1 preadipocytes [84]. Its overexpression could induce changes in gene expression causing adipogenic differentiation and cancer development probably by downregulating catenin beta interacting protein 1 CTNNBIP1, which modulates Wnt cell signaling. The Wnt/β-catenin cell signaling is also related to AT development. Another study showed that overexpression of miR-130a increases osteogenic differentiation and attenuates adipogenic differentiation in bone marrow mesenchymal stem cells (BMSCs) by negatively regulating Smurf2 expression and inhibiting PPARγ expression [48].

Yi et al. showed the expression profile of miRNAs involved in adipogenesis; notably miR-146a-3p, miR-4495, miR-4663, miR-6069, and miR-675-3p were related to lipid droplet formation. A few reports have contributed to confirming the participation of DCK6 in adipogenesis and its regulation by microRNAs [92,93]. A study by Hou et al. showed that cell division of CDK6 participates as a negative regulator in the transition to brown adipose tissue [54]. Feng et al. identified miR-107 as a regulator of CDK6 expression. In addition, Ahonen et al. demonstrated that overexpression of miR-107 in preadipocytes and mature adipocytes reduces adipogenesis by downregulating CDK6, as well as glucose uptake and triglyceride synthesis, respectively [85,94]. Other players in adipocyte differentiation have been revealed. Utilizing combined functional and expression analysis, Chen et. al. found that miRNA-143 promoted the transition from the clonal expansion stage to clonal differentiation by regulating the MAP2K5-ERK5 pathway [86,87]. Employing an in vitro model of the preadipocyte cell line 3T3-L1, an overexpression of miR-375 and a positive effect on preadipocyte differentiation were found, possibly through the ERK-PPARγ2-aP2 pathway [88]. In another study, Hilton et al. proposed the involvement of miR-196a in the regulation of body fat distribution by showing its participation in preadipocyte proliferation and extracellular matrix composition [89].

Conversely, miR-128-3p acts as a negative regulator of adipogenesis as it downregulates the *Pparg* and *Sertad2* genes and subsequently inhibits preadipocyte differentiation and favors lipolysis [49]. Some of the main physiological functions of adipose tissue include the capacity to differentiate into several subtypes, such as white adipose tissue, which is characterized by its endocrine function by modulating caloric intake and triglyceride storage through the formation of the lipid droplet, and brown adipose tissue, which is responsible for thermogenic processes through the oxidation of fats and glucose [56]. Mitochondria is involved in the generation of heat in brown and beige adipocytes. Lemecha et al. investigated the involvement of miR-494-3p during adipogenesis and browning and found decreased expression of miR-494-3p, which downregulates the mitochondrial biogenesis and thermogenesis through PPARγ coactivator1-α (PGC1-α) [57]. Another study performed in human multipotent adipose-derived stem (hMADS) cells showed that miR-1908 inhibits adipocyte differentiation by blocking transcription of the adipogenesis master genes PPARγ and C/EBPα [58]. Brown adipocytes are specialized in the dissipation of energy in the form of heat, a function attributed to the high expression of UCP1. Zfp516, Dio2, and Ppargc1a play a role as transcriptional regulatory genes. Accordingly, Alfonso et al. reported that miR-33 inhibition affects such genes involved in BAT differentiation as well as adaptive thermogenesis [90]. Therefore, it is important to further clarify the process of adipogenesis in each of its stages, as well as the regulations exerted by some miRNAs (Figure 1), in order to understand the way they might be involved in metabolic disorders.

## 5. miRNAs Involved in Carcinogenesis

Cancer is currently considered a serious problem for public health because of its high rate of prevalence and death worldwide [2]. In cancer, there is an increase in cellular metabolic processes such as glycolysis, beta-oxidation of fatty acids, among others. We discuss miRNAs associated with AT and adipogenesis ligated to cancer (Table 2). Several genes involved in distinct carcinogenesis processes have been found to be regulated by microRNAs. In cell proliferation promotion, Feng et al. identified miR-107 as a regulator of CDK6 expression [85,94]. The increased expression of miR-375 in breast cancer has been associated with carcinogenesis [95] and contrarily, in colorectal cancer cells, inhibits proliferation by downregulating JAK2/STAT3 and MAP3K8/ERK cell signaling pathways [96]. The expression of miR-196a has been linked to the development of cancer regulating MAPK pathways [97,98,99]. On the other hand, miR-128-3p was found regulating EGFR-MAPK p38 signaling pathway, promoting metastasis of hepatocellular carcinoma cells through SCAMP3 downregulation [100]. Progression of cancer has been shown through miR-494-3p regulating the PTEN/PI3K/AKT pathway of non-small-cell lung and endometrial cancer [101,102]. The increment of miR-1908-3p has been detected in glioblastoma, osteosarcoma, breast cancer tissues, and serum [103,104,105]. Downregulation of miRNAs is also frequently found affecting carcinogenesis. In this sense, miR-33 expression has been found to be diminished in breast cancer [106]. miR-33 is known to be involved in the regulation and balancing of cholesterol metabolism, fatty acid oxidation, and insulin signaling. In gastric cancer, it was found to be downregulated, and its expression was inversely correlated with pathological differentiation and metastasis. Contrastingly, miR-33a-overexpression was shown inhibiting the capability of the cells to proliferate by cell cycle arrest in G1 phase by targeting CDK6, cyclin D1 (CCND1), and serine/threonine kinase PIM-1 [107]. It is probable that different sets of misregulated genes in distinct types of cancer at a certain stage of disease may drive the function of specific miRNAs on genes, influencing the thorough behavior of cells. Interestingly, it has become evident that some genes directing the progression of metabolic disorders are also seen involved in carcinogenesis. Therefore, understanding the mechanisms of microRNAs regulating carcinogenesis and metabolic disorders in each of their stages would give a wider sense of both pathologies and their management (Figure 1).

## 6. Adipose Tissue Dysregulation in Obesity Can Potentiate Cancer Development through miRNA Interplay

Some of the miRNAs involved in AT deregulation and cancer are discussed. Various studies have suggested the connection between obesity and cancer progression. In this regard, the participation of miR130 in cancer development due to its misregulation in obesity has been strongly suggested. Wang et al. found an increase in miR-130b levels in serum in both mouse and human models of obesity, as well as a positive correlation with BMI. Using 3T3-L1 cell line as an adipogenesis model, it was observed that TGF-β increased miR-130b secretion from mature adipocytes. Therefore, miR-130b could represent a potential biomarker of obesity and some related diseases [122]. TGF-β1 is a significant stimulator of tumor invasion and metastasis in many carcinomas. In colorectal cancer, an increase/decrease in TGF-β1 and miR-130b expression was reported. However, in a different manner, TGF-β1 acted through miR-130b to promote integrin α5 expression, resulting in the enhanced migration of (colorectal cancer cells) CRC cells [118]. In agreement, it has been reported that in epithelial ovarian cancer cells, the TGF-β signaling pathway upregulated miR-130b-3p and reduced cytidine monophosphate kinase expression, a protein that plays an important role in the biosynthesis of nucleoside metabolism, DNA repair, and tumor development [123].

Another miRNA found to be involved in both diseases is miR-107. In human Simpson–Golabi–Behmel syndrome adipocytes (SGB), it was found that miR-107 has different effects on preadipocytes or mature adipocytes by regulating molecules with a role in adipogenesis. For example, in preadipocytes, it was observed that overexpression of miR-107 reduces adipogenesis by downregulating CDK6 and its downstream target genes. On the other hand, in mature adipocytes, miR-107 reduces glucose uptake and triglyceride synthesis, thus promoting ectopic fat deposition [85]. A clearer connection was observed in a study in non-small-cell lung cancer, where miR-107 was found to be downregulated while its target, CDK6, was presented up-regulated, favoring adipogenesis and tumor growth [119]. Additional studies would provide a clearer overview of this connection.

In addition to the myriad of microRNAs implicated in adipocyte differentiation and cancer, Ortega et al. reported the upregulation of miR-221, miR-222, and miR-155 in cells and in supernatants [45]. Interestingly, these miRNAs have been linked to inflammation processes mediated by adipogenesis regulatory networks, which are widely related to an increase in body weight [45] and cancer development [124,125]. In accordance to those findings, it has been shown that M2 macrophages induce colorectal cancer cells’ migration and invasion through the liberation of exosomes containing high levels of miR-21-5p and miR-155-5p [125]. In cancer and obesity, an M1 and M2 macrophage polarization is exerted via changes in circulating glucose and fatty acid substrates, lipotoxicity, and tissue hypoxia [126,127]. Additionally, some of the main functions of the mature adipocyte include the sensitive regulation of insulin. The overexpression of certain miRNAs contributes to the progressive appearance of insulin resistance, such as the upregulated miR-502-3p, affecting insulin sensitivity in fat cells [128]. However, in gastric cancer, miR-502-5p was shown to inhibit NRAS/MEK1/ERK1/2 cell signaling pathway [120], suggesting miRNA strand opposite function. The release of free fatty acids (FFAs) by adipose tissue has resulted in the promotion of tumorigenesis by triggering several cell signaling pathways [129,130], working as metabolic fuel for highly proliferating cells [131,132,133]. Unsaturated FA like oleic acid, palmitoleic acid, and docosahexaenoic acid [127] were found increasing cancer risk and promoting cancer progression [134,135]. Exploring microRNAs participating in the regulation of FA and cancer cellular processes would widen the understanding of the interplay between obesity and malignant diseases.

It is suggested that cross-communication may occur between the obese AT and other different tissues in the organism through miRNA systems. For example, exosomal microRNA-34a secreted by adipocytes inhibits M2 macrophage polarization to promote obesity-induced adipose inflammation. The expression of miR-34a in adipose tissues was progressively increased with the development of dietary obesity and was related to obesity-induced glucose intolerance, insulin resistance, and systemic inflammation, as well as a shift in polarization of adipose-resident macrophages from M1 to M2 phenotype via inhibition of Krüppel-like factor 4 (Klf4) expression [121]. The participation of Klf4 is ambiguous in cancer. In cervical cancer, it has been reported as reduced [136], while in breast cancer, Klf4 expression was overexpressed [137]. The participation of several genes, proteins, and miRNAs should be explored in detail, leading to a deeper understanding of the mechanisms implicated that could represent therapeutic targets for the treatment of inflammation-related diseases.

## 7. Participation of Macrophages in AT and Cancer

Macrophages derived from AT inflammation play an important role by changing their phenotype in response to stimulation. Adipocytes promote the recruitment and polarization of macrophages, either to the classical profile (M1) or to the alternative profile (M2), depending on the degree of adipocyte inflammation. Under such an inflammatory environment, M1 macrophages can induce helper lymphocyte type 1(Th1) responses and potent bactericidal and antitumor activities. M1-type macrophages are characterized by the expression of proinflammatory cytokines such as tumor necrosis factor-α (TNF-α), interleukines (IL-1, IL-6, IL-12), and chemokines such as CXCL1-3, CXCL-5, and CXCL8-10. Conversely, M2-type macrophages can promote Th2 responses, tissue repair, regulatory T lymphocytes (Treg) recruitment through the expression of Transforming growth factor (TGF-β), insulin growth factor (IGF), etc., IL-10, IL-1ra, resulting in an anti-inflammatory microenvironment [138,139]. PPARγ has been associated with macrophages’ change from the M1 to M2 phenotype [50]. Additionally, macrophage change from the M1 to M2 phenotype has been linked to cancer development, and macrophage polarization is a characteristic in obese AT. Constant inflammation in adipose tissue is a major contributor to cancer and obesity-associated metabolic complications. Molecular links between lipid-overloaded adipocytes and inflammatory immune cells in obese adipose and in proliferative tissues are important characteristics in several pathologies. Macrophage polarization is an important central axis in AT and cancer, where M1 macrophages present a proinflammatory status, and M2 macrophages perform an anti-inflammatory M2 phenotype [121,140]. Lactagenesis reduced fatty acid synthesis, enhancing cholesterol biosynthesis in macrophages [141], probably altering their phenotype. It has been reported that lactate dehydrogenase B regulates macrophage metabolism in the tumor microenvironment [141]. Systemic inflammation, as well as a shift in polarization of adipose-resident macrophages from the M1 to M2 phenotype via inhibition of Krüppel-like factor 4 (Klf4) expression, leads to obesity-induced glucose intolerance and insulin resistance. Interestingly, Klf4 expression was regulated by miR-34a, a miRNA that progressively increases with the development of dietary obesity [121] and is related to cervical [142] and breast cancer [143].

Evidence suggests that macrophage polarization plays a pivotal role in promoting both obesity and cancer progression. The connection between macrophages, adipocytes, and tumor cells could partially be performed by the lncRNA–miRNA axis, two systems of regulation that can specifically and broadly regulate cell fate. Whether the reported miRNAs and lncRNAs in obesity and cancer can influence each other’s expression in an autocrine and/or paracrine way is a very interesting question to be explored to further understand the connection between different cell types.

## 8. Biogenesis of lncRNAs

The term lncRNAs denotes those transcripts which are greater than 200 nucleotides in length but lack protein-coding capacity to produce proteins. LncRNAs are observed in a wide diversity of organisms, including animals, plants, and yeast; however, they are poorly conserved among different species. For this reason, along with their low level of expression, lncRNAs were initially considered irrelevant; however, at least in terms of their function [144,145], that perspective has changed at present. LncRNAs are localized mainly in the nucleus. In mammals, they are transcribed mainly by RNA polymerases II or III and have 5′-end m7 G caps and 3′-end poly(A) tails processed similarly to mRNAs [14]. Genes of lncRNAs carry introns and exons and perform RNA maturation processes similar to those of mRNAs, such as conventional and alternative splicing; however, they have also been reported to have different transcription and processing pathways, among others, which could be related to the various cellular targets and activities in which they are involved [14,146]. LncRNAs regulate gene expression by interacting with nucleic acids and proteins in diverse cellular processes [134,135,136]. It has also been reported that lncRNAs may also be involved in the regulation of cancer progression, as well as in other diseases [147,148].

Unlike mRNAs, many Pol II-transcribed lncRNAs are inefficiently processed and are retained in the nucleus, whereas others are spliced and exported to the cytoplasm. The lncRNAs that contain one or only a few exons are exported to the cytoplasm by nuclear RNA export factor 1 (NXF1). Some lncRNAs transcribed by dysregulated Pol II, remain on chromatin and, subsequently, are degraded by the nuclear exosome. Numerous lncRNAs with a certain U1 small nuclear RNA (U1 snRNA) binding motif can recruit the U1 small nuclear ribonucleoprotein (U1 snRNP) and, through it, associate with Pol II at various loci. In many lncRNAs, the sequence between the 3′ splice site and the branch point is longer and contains a shorter polypyrimidine tract (PPT) than in mRNAs, which results in inefficient splicing. Sequence motifs in cis and factors in trans coordinately contribute to nuclear localization of lncRNAs. A nuclear retention element (NRE) U1 snRNA-binding site and C-rich motifs can recruit U1 snRNP19 and heterogeneous nuclear ribonucleoprotein K (hnRNPK), respectively, to enhance lncRNA nuclear localization. Other differentially expressed RNA-binding proteins (RBPs), such as peptidylprolyl isomerase E (PPIE), inhibit splicing of groups of lncRNAs, resulting in their nuclear retention. In the cytoplasm, lncRNAs usually interact with diverse RBPs. Many lncRNAs in the cytoplasm are associated with ribosomes through ‘pseudo’ 5′ untranslated regions (UTRs). Interestingly, ribosome-associated lncRNAs tend to have short half-lives owing to unknown mechanisms, and several lncRNAs are sorted into mitochondria by unknown mechanisms. For example, the RNA component of mitochondrial RNA-processing endoribonuclease (RMRP) lncRNA is recruited to the mitochondria and is stabilized by binding G-rich RNA sequence-binding factor 1 (GRSF1). Some lncRNAs are also found in other organelles, such as exosomes, probably by forming lncRNA–RBP complexes [14]. Importantly, increasing evidence has demonstrated that lncRNA-RBP interactions play a vital role in cancer progression [149]. The mechanism of lncRNAs formation is complex and finely regulated. The interaction of lncRNAs with nucleic acids can function as ceRNAs (competitive endogenous RNAs), generating a web of RNA regulation. Additionally, lncRNAs can regulate protein function by their direct interaction [150]. DNA damage leads to increase in p53 protein levels and stability through phosphorylation and acetylation impeding MDM2 binding to p53 for its degradation. It has been documented that lncRNAs can also participate in p53 stability [151]. DINO (Damage Induced Noncoding RNA) is triggered by DNA damage in a p53-dependent manner. In this regard, DINO is required for the p53-mediated phenotypes, including cell cycle arrest and apoptosis. DINO interacts with and stabilizes p53 which in turn transactivates downstream targets including DINO itself [151]. Therefore, lncRNAs expression and regulation in obesity could be aberrantly modulated, thus contributing to cancer development.

## 9. lncRNAs Regulate miRNA–mRNAs Axis in Obesity

LncRNAs are commonly dysregulated in multiple tumors [152]. An example is lncRNA HAND2-AS1, an antisense strand transcribed near neural crest derivatives and the heart and found twice in chromosome 4q33-34, frequently downregulated in endometrioid endometrial carcinoma (EEC) tissues [153]. Pseudogenes have been found functioning as a competitive endogenous RNA (ceRNA), which can sequester the microRNAs from binding to their target genes [154,155]. LncRNA HAND2-AS1 functions as a ceRNA in cervical cancer by sponging miR-330-5p, regulating cell proliferation, migration, and invasion through LDOC1 modulation [156].

In this work, we determined in silico the lncRNAs that potentially regulate miRNAs associated with cancer and obesity. Seven and eight miRNAs were up- and downregulated, respectively, Figure 2A. A contrasted profile and function were observed for miR-143; the probable network of regulation differs based on the miRNAs listed in miRNet 2.0, Figure 2B,C. The network presented diverse numbers of lncRNAs interacting with miRNAs; eight miRNAs upregulated associated with cancer and obesity can be regulated by 44 lncRNAs. It was shown that miR-107 interacts with 10 lncRNAs, and miR-214 interacts with 7, Figure 2B, while miR-122-3p, miR-196a-5p, miR-494-3p, miR-1908-5p, miR-378a-3p, and miR-143-3p interact with 4 lncRNAs, respectively. Downregulated miRNAs are responsible for the regulation of 49 lncRNAs. miR-128-3p interacts with 11 lncRNAs, miR-34a-5p interacts with 6, while miR-130a-3p, miR-130b-3p, miR-31-5p, and miR-143-3p interact with 5 lncRNAs, Figure 2C. The miRNAs, miR-223-3p, miR-30a-5p, and miR-30e-5p can be regulated by 4 lncRNAs. Members from the families of miR-30 and miR-130 are linked to cancer and obesity. Interestingly, from the networks of miRNAs, lncRNAs KCNQ1OT1 and HCG18 were common to up- and downregulated miRNAs associated with cancer and obesity, Figure 2A–C. However, several lncRNAs have been reported to be involved in cancer, and only few have been reported in obesity.

Several lncRNAs have been found misregulated in breast cancer tissues, highlighting the diversity of genes that can contribute to disease progression. For instance, lncRNAs FGD5-AS1, NEAT1, KCNQ1OT1, HCG18, TTN-AS1, HCG18, OIP5-AS1, and CCDC18-AS1 have been detected as upregulated [157,158,159,160,161,162,163], while a downregulation was reported for miR29B2CHG and GAS5 [164,165]. In lung carcinomas, differential expression of lncRNAs has been observed depending on the cancer type. Representative cases are SLC9A3-AS1, reported upregulated; LINC01554 and GABPB1 reported downregulated in NSCLC tissues [166,167,168], while RBPMS-AS1 expression was found to be diminished in lung adenocarcinoma [169]. Other studies have shown the reduced expression of LINC00294 in colorectal carcinoma tissues [170], while SLFNL1-AS1, KCNQ1OT1, NEAT1, XIST, AC016876.2, AC026362.1 were found to be downregulated in colon cancer cell lines [171]. It should be noticed that not all lncRNAs were analyzed in the same types of samples. Nevertheless, their expression was found to be specific, and therefore, these lncRNAs could be targets of further studies as potential biomarkers. Continuing our exploration of the role of lncRNAs on cancer, recently, it has been reported that PAX8-AS1 expression was not differing between normal tissue and cervical cancer [172]; however, PCBP1-AS1 expression was increased in cervical cancer tissues as well as associated with the tumor stage, TNM, and invasion [173]. Furthermore, highlighting their role in cancer, a high expression of MIR181A1HG in self-renewing mesenchymal stromal cells (MSCs) has been shown, and it has been linked with lineage progression of committed osteoblast precursors [174]. Therefore, the limited expression of lncRNAs on sections of different organs and carcinomas supports the complexity of cancer and justifies the study of their participation in gene regulation to further enlighten the molecular mechanisms of carcinogenesis.

Despite their importance in different pathologies, information on lncRNAs in obesity and cancer is limited, especially in the former. It was previously reported that lncRNAs Meg3, Malat1, Neat1, and Kcnq1ot1 were found increased in obese mice [175]. Other lncRNAs, such as Mist, lincIRS2, lncRNA-p5549, H19, GAS5 and SNHG9, and OIP5-AS1 have been shown downregulated in adipose tissues of obese humans [20,176], while Meg3, Plnc1, Blnc1, AC092834.1, TINCR, and PVT1 were upregulated in the obese subjects [20].

## 10. Conclusions

Obesity and related diseases such as cancer are a concern of global interest. The regulation in the expression of genes involved in cellular and molecular processes of the cells, as well as the essential target molecules for these processes to occur, are vital to maintaining the homeostasis of the organism. In the present review, we present evidence highlighting the dysregulation of molecules such as miRNAs and lnRNAs involved in the obesity-related complications and in some types of cancers, as well as in the possible connection between these two pathologies. Several cell-signaling pathways are regulated by miRNAs and lncRNAs. In particular, miR-130, miR-4663, miR-375, miR-494-3p, and miR-1908 regulate PPARγ cell signaling; on the other hand, miR-143, miR-375, miR-196, and miR-128-3p regulate MAPK cell signaling. The network of regulation is complex, involving more than two types of molecules. The miRNAs misregulated in obesity and cancer are modulated by 44 upregulated and 49 downregulated lncRNAs. From these, miR-375, miR-494-3p, miR-1908, and miR-196 are regulated by 1, 4, 4, and 4 lncRNAs, respectively, and these miRNAs and lncRNAs are involved in PPARγ cell signaling regulation by distinct mRNAs and proteins. In contrast, miR-130 was found to be downregulated in obesity modulating 5 lncRNAs, as well as a player in PPARγ cell signaling. Additionally, miR-128-3p and miR-143, are downregulated in obesity and cancer and interact with 5 and 4 lncRNAs, respectively, that are related to MAPK cell signaling modulation. It should be noted that miR-143-3p exhibits both up- and downregulation in obesity by different studies. The cumulative findings reviewed here suggest that the delicate balance between the expression of up- and downregulated miRNAs and lncRNAs can dictate the fate of developing cancer associated with obesity. There is still a long way to go to understand how this dysregulation can be modified and provide a prompt diagnosis and better prognosis in these diseases. A long way to understand the mechanisms of miRNA–lncRNA dysregulation is evident, and further work is necessary in order to benefit from these molecules as targets to fight cancer development and progression, as well as to utilize them like markers of prognosis and diagnosis.

## Figures and Tables

**Figure 1 ncrna-10-00024-f001:**
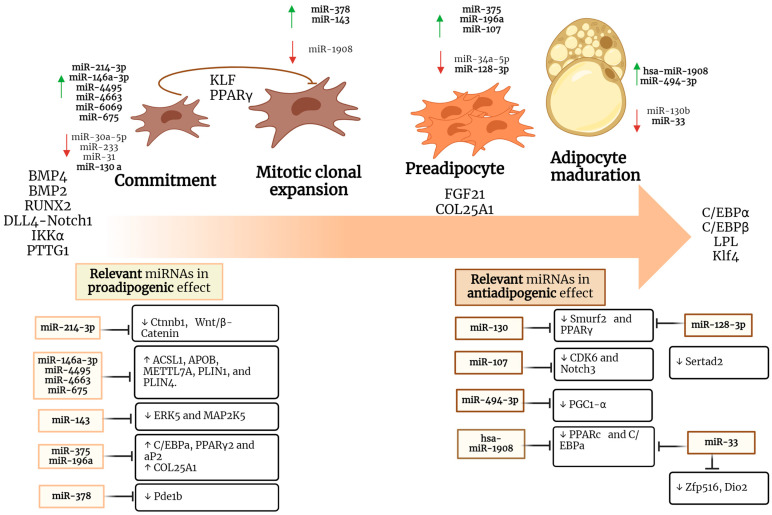
Adipogenesis process and its regulation mediated by miRNAs. miRNAs involved in adipogenesis. The development of fat cells from mesenchymal stem cells involves a complex process regulated by several genetic factors and mechanisms. The initiation of adipogenesis is mediated by intracellular signals that activate the differentiation of MSCs into the adipogenic lineage. During adipogenic differentiation, specific genes that are involved in fat cell formation, such as PPARγ (peroxisome proliferator-activated receptor gamma) and C/EBP (CCAAT regulatory element binding protein) are activated. This process is also mediated by specific miRNAs, which are upregulated or inhibited, promoting changes in the adipogenic response.

**Figure 2 ncrna-10-00024-f002:**
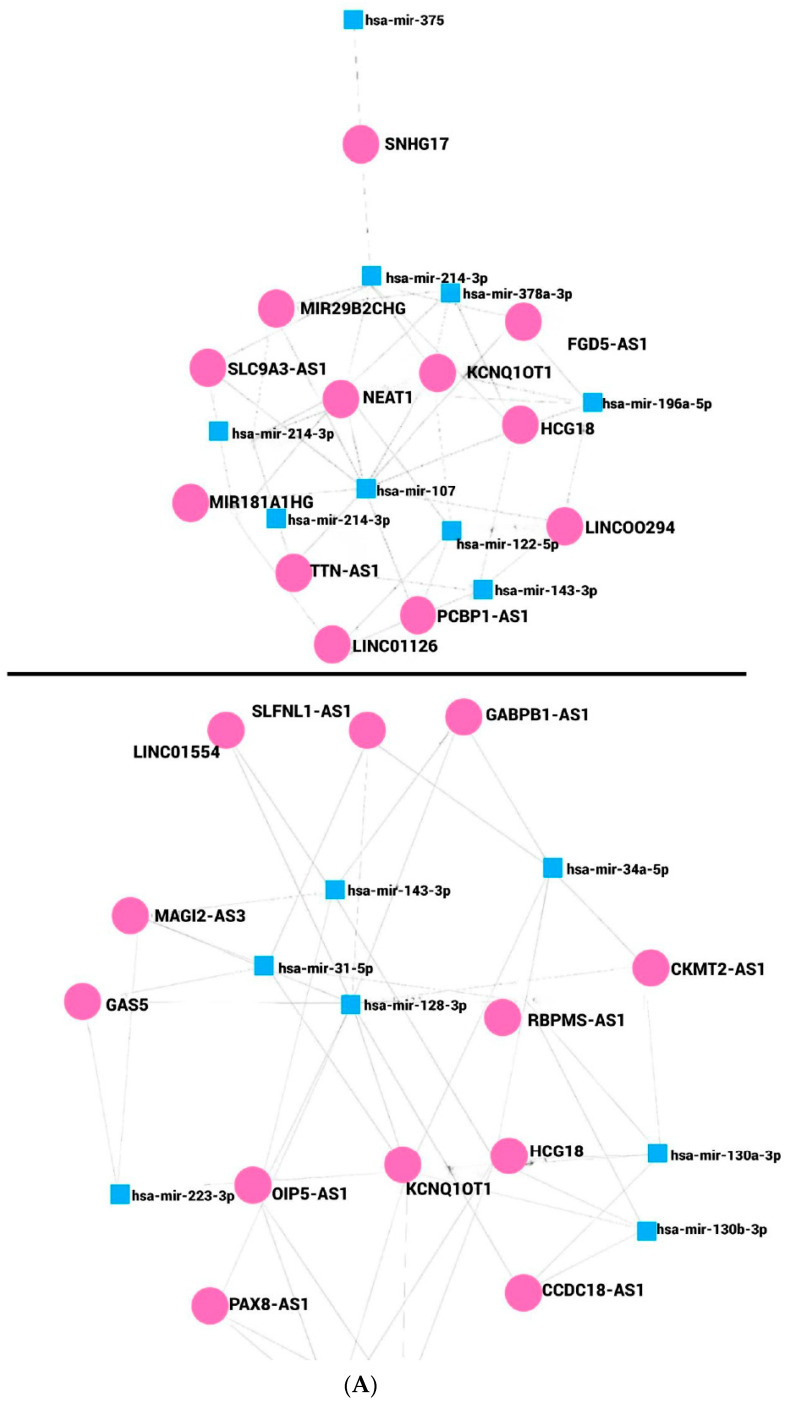
Network of miRNAs and lncRNAs in cancer and obesity. (**A**) Network regulation between miRNAs and lncRNAs. The interconnection between miRNA–miRNA and miRNA–lncRNA is illustrated. (**B**) Interaction between upregulated miRNAs and lncRNAs. The association between upregulated miRNAs and lncRNAs is illustrated. (**C**) Interaction between downregulated miRNAs and lncRNAs. The association between downregulated miRNAs and lncRNAs is illustrated.

**Table 1 ncrna-10-00024-t001:** Summary of relevant miRNAs involved in adipogenesis process.

miRNA	Effect	Targets	Model Study	Reference
miR-214-3p	Proadipogenic	↓ Ctnnb1, Wnt/β-Catenin	3T3-L1 cells	[84]
miR-130a	Antiadipogenic	↓ Smurf2, PPARγ	Mouse BMSC	[48]
miR-146a-3p, miR-4495, miR-4663, miR-6069, miR-675-3p	Proadipogenic	↑ ACSL1, APOB, METTL7A, PLIN1, PLIN4. A	Human MSCs	[52]
miR-107	Antiadipogenic	↓ CDK6, Notch3	Human SGBS	[85]
miR-143	Proadipogenic	↓ ERK5, MAP2K5	Human White preadipocytes	[86,87]
miR-375	Proadipogenic	↑ C/EBPa, PPARγ2, aP2	3T3-L1 cells	[88]
miR-196a	Proadipogenic	↑ COL25A1	ASAT biopsies	[89]
miR-128-3p	Antiadipogenic	↓ Pparg, Sertad2	3T3-L1 cells	[49]
miR-494-3p	Antiadipogenic	↓ PGC1-α	3T3-L1 cells	[57]
miR-1908	Antiadipogenic	↓ PPARc, C/EBPa	hMADS cells	[58]
miR-33	Antiadipogenic	↓ Zfp516, Dio2, and Ppargc1a	10T1/2 cells	[90]
miR-378	Proadipogenic	↓ Pde1b	C3H10T1/2 cells	[91]

**Table 2 ncrna-10-00024-t002:** Summary of deregulation of miRNAs relevant in cancer development.

miRNA	Dysregulation	Mediator Affected	Cancer Related	Reference
miR-214-3p	Overexpressed	↓ CTNNBIP1	Breast cancer	[108]
		↑ Nanog, EpCAM	Cancer stem-like cells	[109]
		↑ Oct-4, Nanog, Sox-2, CD133, and EpCAM	Lung adenocarcinoma	[109]
miR130a	Underexpressed	↓ PPARγ	Non-small cell lung cancer	[50]
miR-146a-3p	Overexpressed	↓ IRAK1 and TRAF6	Cervical Cancer	[110,111]
	Underexpressed	↑ PTTG1	Bladder cancer	[112]
miR-4663	Overexpressed	↓ Paxs ↑ PPARs	Esophageal carcinoma	[113]
miR 675	Overexpressed	↓ DMTF1	Colorectal cancer	[114]
mirR-107	Overexpressed	↓ NEDD9	Breast cancer	[55]
miR-143-3p	Underexpressed	↓ Ras/Raf/MEK/ERK	Wilms’ tumor	[115]
	Underexpressed	↓ KRAS	Pancreatic ductal adenocarcinoma	[116]
	Underexpressed	↓ N-RAS	Glioma	[117]
miR-375	Overexpressed	↓ FOXO1	Breast cancer	[95]
	Underexpressed	↓ JAK2/STAT3 and MAP3K8/ERK	Colorectal cancer	[96]
miR-196a	Overexpressed	↓ Bram1	Renal cancer	[99]
miR-128-3p	Underexpressed	↓ Pparg and Sertad2 ↑ EGFR-MAPK p38	Hepatocellular carcinoma	[49,100]
miR-494-3p	Overexpressed	↓ PGC1-α ↑ PTEN/PI3K/AKT	Cancer cancer non-small cell lung and endometrial	[57,101]
miR-1908	Overexpressed	↓ PPARγ and C/EBPα	Glioblastoma, osteosarcoma and breast cancer tissues	[103,104,105]
miR-33	Underexpressed	↓ CCND1 and CDK6	Breast cancer	[106]
miR-130b	Underexpressed	↑ TGF-β1 and integrin α5	Epithelial ovarian cancer cells	[118]
miR-107	Underexpressed	↑ CDK6	Lung cancer	[119]
miR-502-5p	Overexpressed	↓ NRAS/MEK1/ERK1/2	Gastric cancer	[120]
miR-34a	Overexpressed	↓ Klf4	In cervical cancer	[121]

## Data Availability

All data generated and analyzed during this study are included in this published article.

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
