# Peer review of "miRNAs as Interconnectors between Obesity and Cancer"

_ncrna, 2024, doi:10.3390/ncrna10020024_

Round 1
Reviewer 1 Report
Comments and Suggestions for Authors
This review article written by Sánchez and colleagues titled “miRNAs as interconnectors between obesity and cancer” try to elucidate the role of miRNAs and IncRNAs as mediators of obesity induced cancer in human beings. The authors try to address an important and interesting issue. However, the way this review article is structured has significant flaws. This review article reads like a research paper (not like a review article) and needs lot of work to improve it. For instance, typical review articles use the abstract to outline the main points to be addressed in the body of the articles, not to present what read like experimental results as expected in research articles. Authors need to revise this review article before accepting for publication. Authors need to address following issues accepting for publication.
1. In the abstract, lines 20-28 (“In obesity…..promising markers”) need to be moved to the body of the article and be replaced with a formal outline of the review article. Something like “in this review, we will discuss X, Y, and Z, will help readers preview the article. The last sentence (beginning in line 26) needs clarification.
2. The focus of this review article is miRNAs and lncRNAs which are barely mentioned in the introduction (just one sentence, line 69-71). More background/introductory information is needed on these molecules and to preface/preview how they potentially interlink with obesity.
3. Readability issues were noted throughout the manuscript, with grammatical errors and typos, which significantly affect the readability of this article. For instance, line 114…(MSCs, as pluripotent adult stem cells have can be isolated from many tissues, including muscle tissue, tendon, synovial membrane….) and line 386... “miRNAs downregulated regulated are responsible for the regulation of 49 lncRNAs.” The ideas are there but the typos and grammatical errors obscured them. Similar issues recur throughout the manuscript.
4. Section 2.1 reads like the biogenesis of miRNA and not as what it was designated for. If biogenesis of miRNAs was not the intention, it will help readers to have that before the “Function and molecular action of MicroRNAs”
5. The idea that MSCs can produce “preadipocytes” by the action of “adipogenic inducers” (line 119-120) does not make sense. The references (45 and 47) provided do not contain the data involving BMP4 and Wnt family protein in such MSCs physiology. As stem cells, MSCs can be differentiated into adipocytes using adipogenic factors and therefore MSCs can be likened to “preadipocyte”. As such, these “preadipocytes” cannot produce “preadipocytes” by the action of adipogenic factors which are supposed to drive differentiation into adipocytes. The authors need to double-check that statement.
6. Section 2.2 “Adipogenesis is linked to cancer development via miRNAs regulation” appears to address a lot at once without a clear structure. It is very long and may cause readers to lose focus. It is recommended that the authors divide that section accordingly, by first addressing “the adipogenic process”, then “adipogenesis and cancer” and lastly “the link of miRNAs to adipogenesis and cancer”
7. Table 1 (page 4) is not clear and difficult to follow. Simplifying this table will help readers to understand it easily.
8. Section 2.3 “Adipose tissue dysregulation in obesity can potentiate cancer development through miRNA interplay” is also long and may benefit readers to be sub-divided into miRNAs mediate the dysregulation of adipose tissue in obesity and how that drives cancer development.
9. From section 2.5 onward, figures should be parenthesized.
10. “Authors contribution” section (line 459-470) suggests that the article was indeed treated as a research article not as a review article. This reviewer wonders what the contribution to “Methodology, investigation, formal analysis, data curation, validation” refers to in this article if this is a review article.
Comments on the Quality of English Language
English language quality should be improved before publication.
Author Response
Dear reviewer thank you for the suggestions and comments.
- In the abstract, lines 20-28 (“In obesity…..promising markers”) need to be moved to the body of the article and be replaced with a formal outline of the review article. Something like “in this review, we will discuss X, Y, and Z, will help readers preview the article. The last sentence (beginning in line 26) needs clarification.
RESPONSE: lines 20-28 were changed as suggested. Line26 was clarified.
- The focus of this review article is miRNAs and lncRNAs which are barely mentioned in the introduction (just one sentence, line 69-71). More background/introductory information is needed on these molecules and to preface/preview how they potentially interlink with obesity.
Response: information of miRNAs and lncRNAs was added in the introduction.
- Readability issues were noted throughout the manuscript, with grammatical errors and typos, which significantly affect the readability of this article. For instance, line 114…(MSCs, as pluripotent adult stem cells have can be isolated from many tissues, including muscle tissue, tendon, synovial membrane….) and line 386... “miRNAs downregulated regulated are responsible for the regulation of 49 lncRNAs.” The ideas are there but the typos and grammatical errors obscured them. Similar issues recur throughout the manuscript.
Response: The manuscript was check it and corrected.
- Section 2.1 reads like the biogenesis of miRNA and not as what it was designated for. If biogenesis of miRNAs was not the intention, it will help readers to have that before the “Function and molecular action of MicroRNAs”
Response: The title of the section was changed to biogenesis of miRNA
- The idea that MSCs can produce “preadipocytes” by the action of “adipogenic inducers” (line 119-120) does not make sense. The references (45 and 47) provided do not contain the data involving BMP4 and Wnt family protein in such MSCs physiology. As stem cells, MSCs can be differentiated into adipocytes using adipogenic factors and therefore MSCs can be likened to “preadipocyte”. As such, these “preadipocytes” cannot produce “preadipocytes” by the action of adipogenic factors which are supposed to drive differentiation into adipocytes. The authors need to double-check that statement.
Response: MSCs can differentiate into adipocytes by adipogenic inducers. References were added to support BMP and Wnt proteins in MSCs physiology.
- Section 2.2 “Adipogenesis is linked to cancer development via miRNAs regulation” appears to address a lot at once without a clear structure. It is very long and may cause readers to lose focus. It is recommended that the authors divide that section accordingly, by first addressing “the adipogenic process”, then “adipogenesis and cancer” and lastly “the link of miRNAs to adipogenesis and cancer”
Response: Section were organized as suggested.
- Table 1 (page 4) is not clear and difficult to follow. Simplifying this table will help readers to understand it easily.
Response: Table 1 was split in two to clarified the information.
- Section 2.3 “Adipose tissue dysregulation in obesity can potentiate cancer development through miRNA interplay” is also long and may benefit readers to be sub-divided into miRNAs mediate the dysregulation of adipose tissue in obesity and how that drives cancer development.
Response: Section were organized as suggested.
- From section 2.5 onward, figures should be parenthesized.
A parenthesis was added to all figures.
- “Authors contribution” section (line 459-470) suggests that the article was indeed treated as a research article not as a review article. This reviewer wonders what the contribution to “Methodology, investigation, formal analysis, data curation, validation” refers to in this article if this is a review article.
Response: The structured of the paper was changed.
Reviewer 2 Report
Comments and Suggestions for Authors
The authors of the review titled "miRNAs as Interconnectors between Obesity and Cancer" have chosen a pertinent topic. While they present some valuable insights and have done a comprehensive literature review, the organization and relevance of the information need improvement. For publication consideration, an extensive revision is necessary. Key areas for revision include:
- The second paragraph on page 2 lacks relevance to the overall theme of the review, please explain how SNP can be involved in the process.
- The title "Results and Discussion" is not appropriate for a review article. Please consider revising it.
- The content under the subheading "Function and Molecular Action of MicroRNAs" does not align with the title of the subheading.
- Revisit the manuscript for editing, specifically on page 3, lines 114-116, which discuss the multilineage development potential of MSCs from various tissues.
- Discuss the types of autocrine and paracrine communication, and how these are regulated, in the context of the review.
““Autocrine and paracrine communication as well as morphological changes and gene expression result in the transformation of preadipocytes into mature adipocytes [44-46].”
- Explain the relevance of MSCs in cancer induction or progression, particularly their role in NSCLC via the AMPK signaling pathway, as mentioned on page 5, lines 166-169. Information about miR/lncRNA and associated pathways will enhance the information.
- The table presented is unclear and lacks organization and a legend, making it difficult to extract meaningful information.
- Clarify the connection of RASopathies to the scope of the review.
- Elaborate on how LncRNAs regulate gene expression through interactions with nucleic acids and proteins.
Comments on the Quality of English Language
Please revise the manuscript, specifically focusing on the section on page 3, lines 114-116, where it states, 'MSCs, as pluripotent adult stem cells, can be isolated from various tissues such as muscle, tendon, synovial membrane, dental pulp, skin, lung, placenta, and adipose tissue. This diversity in tissue origin endows these cells with the capability for multilineage development.'
Author Response
Dear reviewer thank you for the suggestions and comments.
1. The second paragraph on page 2 lacks relevance to the overall theme of the review, please explain how SNP can be involved in the process.
Response: The paragraph was erased.
2. The title "Results and Discussion" is not appropriate for a review article. Please consider revising it.
Response: The structure of the paper was changed.
3. The content under the subheading "Function and Molecular Action of MicroRNAs" does not align with the title of the subheading.
Response: The subheading "Function and Molecular Action of MicroRNAs" was changed for biogenesis.
4. Revisit the manuscript for editing, specifically on page 3, lines 114-116, which discuss the multilineage development potential of MSCs from various tissues.
Response: Lines were check it.
5. Discuss the types of autocrine and paracrine communication, and how these are regulated, in the context of the review.
““Autocrine and paracrine communication as well as morphological changes and gene expression result in the transformation of preadipocytes into mature adipocytes [44-46].”
Response: Information suggested was added.
6. Explain the relevance of MSCs in cancer induction or progression, particularly their role in NSCLC via the AMPK signaling pathway, as mentioned on page 5, lines 166-169. Information about miR/lncRNA and associated pathways will enhance the information.
Response: Information suggested was added
7. The table presented is unclear and lacks organization and a legend, making it difficult to extract meaningful information.
Response: Table 1 was split in two to clarified the information.
8. Clarify the connection of RASopathies to the scope of the review.
Response: The information was removed.
9. Elaborate on how LncRNAs regulate gene expression through interactions with nucleic acids and proteins.
Response: Information suggested was added
Round 2
Reviewer 1 Report
Comments and Suggestions for Authors
Authors have addressed the issues I have pointed out in the review-article and this revised version is acceptable for publication.
Author Response
Dear reviewer, thank you very much for your time, observations and suggestions.
Reviewer 2 Report
Comments and Suggestions for Authors
The authors have done an extensive revision of the study and have addressed most of the comments, but the review still needs to be revised and needs extensive language editing. Here are some examples:
Abstract:
1. For instance, miR-375, miR-494-3p, miR-1908, miR-196 were found interacting with 1, 4, 4 and 4 lncRNAs, respectively which are involved in PPARÉ£ cell signaling regulation--
Other than the language editing consider writing like: miR-375, miR-494-3p, miR-1908, and miR-196--. Please add the information about what is: 1,4,4 and 4 lncRNA, and why they are important to consider.
2. “In contrast, miR-130 was found downregulated in obesity has been reported modulating 5 lncRNAs”
The purpose of the review is to give detailed information, modulating 5 lncRNA does not serve the purpose, adding what 5 lncRNA is and why they may in important is suggested.
3. A few reports have contributing to confirm the participation of DCK6 in adipogenesis and its regulation by microRNAs. A study showed that cell division of CDK6 participates as a negative regulator in the transition to brown adipose tissue [76].
A few reports: which?
A study showed: A study by ----- showed-----
Comments on the Quality of English Language
Need language editing.
Author Response
Dear reviewer we try to address point by point.
Abstract:
- For instance, miR-375, miR-494-3p, miR-1908, miR-196 were found interacting with 1, 4, 4 and 4 lncRNAs, respectively which are involved in PPARÉ£ cell signaling regulation--
Other than the language editing consider writing like: miR-375, miR-494-3p, miR-1908, and miR-196--. Please add the information about what is: 1,4,4 and 4 lncRNA, and why they are important to consider.
Response: The suggestion was added.
- “In contrast, miR-130 was found downregulated in obesity has been reported modulating 5 lncRNAs”
The purpose of the review is to give detailed information, modulating 5 lncRNA does not serve the purpose, adding what 5 lncRNA is and why they may in important is suggested.
Response: We agree with the observation and we added the cell signaling regulated by these lncRNAs.
- A few reports have contributing to confirm the participation of DCK6 in adipogenesis and its regulation by microRNAs. A study showed that cell division of CDK6 participates as a negative regulator in the transition to brown adipose tissue [76].
A few reports: which?
Response: Additionally, references were added as suggested.
A study showed: A study by ----- showed-----
Response: The grammatical structure suggested was added.
Language editing was check it by 2 colleagues.
Round 3
Reviewer 2 Report
Comments and Suggestions for Authors
The revised manuscript is in good shape.
Comments on the Quality of English Language
Language editing will be helpful.
Author Response
Thanks for your review.